# Histology Classification Highlights Differences in Efficacy of S-1 versus Capecitabine, in Combination with Cisplatin, for HER2-Negative Unresectable Advanced or Recurrent Gastric Cancer with Measurable Disease

**DOI:** 10.3390/cancers14225673

**Published:** 2022-11-18

**Authors:** Hisato Kawakami, Kazuhiro Nishikawa, Toshio Shimokawa, Kazumasa Fujitani, Shigeyuki Tamura, Shunji Endo, Michiya Kobayashi, Junji Kawada, Yukinori Kurokawa, Akira Tsuburaya, Takaki Yoshikawa, Junichi Sakamoto, Taroh Satoh

**Affiliations:** 1Department of Medical Oncology, Kindai University Faculty of Medicine, Osaka-Sayama, Osaka 589-8511, Japan; 2Department of Surgery, Sakai City Medical Center, Sakai, Osaka 593-8304, Japan; 3Clinical Study Support Center, Wakayama Medical University, Wakayama, Wakayama 641-0012, Japan; 4Department of Surgery, Osaka Prefectural General Medical Center, Osaka, Osaka 558-8558, Japan; 5Department of Surgery, Yao Municipal Hospital, Yao, Osaka 581-0069, Japan; 6Department of Digestive Surgery, Kawasaki Medical School Hospital, Kurashiki, Okayama 701-0192, Japan; 7Cancer Treatment Center, Kochi Medical School Hospital, Nangoku, Kochi 783-8505, Japan; 8Department of Gastroenterological Surgery, Osaka University Graduate School of Medicine, Suita, Osaka 565-0871, Japan; 9Department of Surgery, Ozawa Hospital, Odawara, Kanagawa 250-0012, Japan; 10Department of Gastric Surgery, National Cancer Center Hospital, Tokyo 104-0045, Japan; 11Tokai Central Hospital, Kakamigahara, Gifu 504-8601, Japan; 12Palliative and Supportive Care Center, Osaka University Hospital, Suita, Osaka 565-0871, Japan

**Keywords:** advanced gastric cancer, recurrent gastric cancer, S-1, capecitabine, cisplatin

## Abstract

**Simple Summary:**

There is no clear preference between S-1 and capecitabine in combination with platinum agent as first-line therapy for patients with HER2-negative unresectable advanced or recurrent gastric cancer (GC) with measurable disease. Although a distinguishing use of S-1 versus capecitabine based on histological classification has been studied, the present integrated analysis, using 162 individual participant data of GC patients with measurable disease, is the first to show that S-1 plus cisplatin (SP) achieves deeper tumor shrinkage than capecitabine plus cisplatin (XP), leading a longer overall survival although no differences in response rate or progression-free survival in differentiated GC. On the other hand, in undifferentiated GC, SP consistently showed better clinical results than XP. These findings thus have implications for the choice of oral fluoropyrimidine in the era of first line therapy in combination with oxaliplatin and immune checkpoint inhibitor.

**Abstract:**

It has been suggested that the therapeutic efficacy of S-1 + cisplatin (SP) and capecitabine + cisplatin (XP) may differ depending on the histology of the tumor, but no clear evidence exists. Individual participant data were obtained from three randomized phase II trials in which such patients received either SP (S-1 [40–60 mg twice daily for 21 days] plus cisplatin [60 mg/m^2^ on day 8], every 5 weeks) or XP (capecitabine [1000 mg/m^2^ twice daily for 14 days] plus cisplatin [80 mg/m^2^ on day 1], every 3 weeks). A total of 162 patients were included, with 79 patients in the SP arm and 83 patients in the XP arm. Although there was also no difference between arms in ORR according to histological classification, differentiated tumors showed a significantly better OS (but not PFS) for SP versus XP that was associated with a deeper tumor shrinkage. Undifferentiated tumors showed a consistently better OS, and PFS for SP versus XP, likely because cases without tumor shrinkage tended to be fewer for SP. Our data thus showed that SP was superior to XP in this setting, but there were qualitative differences in therapeutic efficacy dependent on tumor histology.

## 1. Introduction

Gastric cancer remains one of the most common and fatal cancers worldwide, with GLOBOCAN data for 2020 showing it to be the fifth most frequent and fourth most deadly cancer, with an estimated 769,000 deaths [1]. Individuals with newly diagnosed gastric cancer often present with unresectable or metastatic disease. For these patients, as well as the small number of cases of recurrence after radical surgery, cytotoxic chemotherapy is administered to alleviate symptoms, improve quality of life, and prolong survival. Most advanced gastric tumors are negative for human epidermal growth factor receptor 2 (HER2), with the standard first-line chemotherapy for these cases being a two-drug combination of platinum and a fluoropyrimidine, the latter of which is usually oral S-1 or capecitabine in Japan.

S-1 has three components in a molar ratio of 1:0.4:1: tegafur, a prodrug of 5-fluorouracil (5-FU); gimeracil, which selectively inhibits the metabolic degradation of 5-FU; and potassium oteracil, which limits mucosal damage [2]. In Japan, the combination of S-1 and cisplatin (SP) has been regarded as a standard of care for advanced gastric cancer on the basis of the results of the SPIRITS study, in which 5-week cycles of SP showed an overall survival (OS) benefit versus S-1 alone [3]. Capecitabine is a prodrug of 5-FU that is hydrolyzed by carboxylesterase in the liver to 5′-deoxy-5-fluorocytidine (5′-DFCR), which is then converted to 5′-deoxy-5-fluorouridine (5′-DFUR) by cytidine deaminase in the liver and tumor tissue, and 5′-DFUR is finally converted to 5-FU by thymidine phosphorylase in tumor tissue [4]. Capecitabine plus cisplatin (XP) in 3-week cycles has been administered as a standard of care for advanced gastric cancer in several global trials [5,6,7] because of its noninferiority to 5-FU plus cisplatin [8]. Both SP and XP are listed in the Japanese guidelines as first-line treatments for advanced gastric cancer [9]. Given the difference in design and concept for S-1 and capecitabine, however, direct comparison of these two oral fluoropyrimidine drugs in combination with cisplatin is desirable to discriminate the use of SP versus XP.

As far as we are aware, three randomized phase II trials have prospectively compared SP and XP for HER2-negative unresectable advanced or recurrent gastric cancer in Japan: the HERBIS-2 [10] and HERBIS-4A [11] trials performed by the Osaka Gastrointestinal Cancer Chemotherapy Study Group (OGSG) and the XParTS II trial [12] conducted by the Epidemiological and Clinical Research Information Network (ECRIN). However, these trials reported different results. In the pooled analysis of HERBIS-2 and HERBIS-4A, SP showed a longer progression-free survival (median PFS: 6.4 vs. 5.1 months, with a hazard ratio [HR] of 0.666, *p* = 0.062), OS (median of 14.8 vs. 10.6 months, with an HR of 0.695, *p* = 0.099), and time to treatment failure (median TTF: 4.6 vs. 3.6 months, with an HR of 0.668, *p* = 0.045) compared with XP, whereas there was no difference in overall response rate (ORR: 54.5% vs. 51.1%, {respectively, *p* = 1.000) [10]. On the other hand, the XParTS II study found that the ORR was significantly lower for SP than for XP (42.4% vs. 69.4%, *p* = 0.0237), with a comparable outcome of SP versus XP with regard to PFS (5.6 vs. 5.1 months, respectively, with an HR of 1.126, *p* = 0.5626) and OS (13.5 vs. 12.6 months, with an HR of 0.942, *p* = 0.7769) [12]. Exploratory analysis based on histological classification did not detect any difference in efficacy between treatments in any of these trials.

It has therefore remained unknown whether S-1 or capecitabine is more suitable as a first-line treatment for this setting. Even though the two drugs are intrinsically different, the appropriate use of each remains unclear. One reason for this ambiguity is the dearth of significant differences in the clinical trials as a result of their limited sample sizes. The integration of individual participant data (IPD) would allow analysis of a larger number of patients. Given that the regimens administered in each study are identical, and that the eligibility and exclusion criteria as well as the enrollment periods (2012–2014, 2011–2016, and 2011–2013 for HERBIS-2, HERBIS-4A, and XParTS II, respectively) of the studies largely overlap, we have now performed an integrated analysis of these three trials. In this analysis, we here reveal qualitative differences in therapeutic efficacy between SP and XP for HER2-negative unresectable advanced or recurrent gastric cancer with a measurable lesion that are dependent on histology.

## 2. Patients and Methods

### 2.1. Study Design and Treatment

The analysis was designed in 2020. Both OGSG and ECRIN gave their approval according to a formal protocol. Each of the selected randomized phase II trials compared SP versus XP with the same dosing, and the treatment methods were identical. In all three trials, patients were randomly assigned to receive either SP (S-1 at 40–60 mg twice daily for 21 days plus cisplatin at 60 mg/m^2^ on day 8) every 5 weeks or XP (capecitabine at 1000 mg/m^2^ twice daily for 14 days plus cisplatin at 80 mg/m^2^ on day 1) every 3 weeks. Each treatment was continued until disease progression or the development of intolerable toxicity. Inclusion and exclusion criteria for patient enrollment (see below) were described previously [10,11,12].

We first verified the integrity of IPD from the HERBIS-2, HERBIS-4A, and XParTS II trials. All clinical data were extracted and held centrally at the data center of OGSG for HERBIS-2 and HERBIS-4A and at that of ECRIN for XParTS II. This analysis was conducted according to the Declaration of Helsinki. All patients in the three trials provided written consent after being informed about the purpose and investigational nature of the respective studies. The institutional review boards or ethics committees of all participating centers reviewed and approved the protocol for the present analysis.

### 2.2. Patients

Patients with histologically confirmed HER2-negative unresectable advanced or recurrent gastric cancer were eligible for all three trials. HER2 positivity was defined as 3+ staining by immunohistochemistry or as HER2 gene amplification (HER2:CEP17 signal ratio of ≥2.0) as detected by in situ hybridization. For HERBIS-4A, patients were limited to those who were naïve to systemic chemotherapy [11], whereas those with disease recurrence during or within 6 months after the completion of adjuvant therapy with S-1 were eligible for HERBIS-2 [10]. For XParTS II, eligibility stipulated no previous chemotherapy or radiotherapy, with the exception of adjuvant chemotherapy if completed >6 months before enrollment [12]. Other eligibility criteria were as follows: age of ≥20 years (and ≤74 years for XParTS II); estimated life expectancy of ≥12 weeks; written informed consent; an Eastern Cooperative Oncology Group performance status of ≤2; and adequate organ function as reflected by a white blood cell count of ≥3000/mm^3^ for HERBIS-2 and -4A, neutrophil count of ≥1500/mm^3^, platelet count of ≥100,000/mm^3^, hemoglobin level of ≥8.0 g/dL, aspartate aminotransferase (AST) and alanine aminotransferase (ALT) levels of ≤100 IU/L (≤150 IU/L in cases with metastasis to the liver) for HERBIS-2 and -4A or of ≤2.5× the upper limit of normal (ULN) at each institution (≤5× in cases with metastasis to the liver) for XParTS II, total bilirubin level of ≤1.50 mg/dL for HERBIS-2 and -4A or of ≤1.5× the ULN at each institution for XParTS II, serum creatinine level of ≤1.20 mg/dL for HERBIS-2 and -4A, and creatinine clearance of ≥60 mL/min (as estimated by the Cockcroft-Gault equation). Exclusion criteria included additional malignancies and serious comorbidities.

### 2.3. Endpoints and Assessments

The primary objective of the present study was to integrate the prospectively collected IPD from the HERBIS-2, HERBIS-4A, and XParTS II trials in order to compare SP with XP and to determine the optimal first-line chemotherapy for patients with HER2-negative unresectable advanced or recurrent gastric cancer. We examined differences in response between SP and XP, focusing on tumor histology and the form of the response such as the distribution of tumor shrinkage and changes in tumor burden, and how they might affect survival.

Tumor responses were analyzed according to Response Evaluation Criteria in Solid Tumors (RECIST) version 1.1 in patients with at least one measurable lesion at baseline, and they were classified as complete response (CR), partial response (PR), stable disease (SD), or progressive disease (PD). The ORR was defined as the proportion of patients with a CR or PR, and the disease control rate (DCR) as the proportion of patients with a CR, PR, or SD. Deepness of response (DpR) was defined as the maximum tumor shrinkage (%) observed throughout treatment.

Tumor histology was based on the Japanese classification of gastric carcinoma [9], with differentiated-type tumors being defined as papillary or tubular adenocarcinoma and undifferentiated-type tumors as poorly differentiated adenocarcinoma, signet ring cell carcinoma, or mucinous adenocarcinoma. Other histology types were designated as “other”.

### 2.4. Statistical Analysis

OS was defined as time from randomization to death, PFS as time from randomization to radiographic progression or death. Time to response (TTR) was defined as the interval between the start of therapy and the first confirmation of a response, and duration of response (DOR) as the time from documentation of a response to that of disease progression. Survival curves were constructed as time-to-event plots with the Kaplan–Meier method. Time-to-event curves were compared with the log-rank test, and HRs were estimated with Cox regression models. The confidence coefficient for confidence intervals (CIs) of median OS, PFS, DOR, TTR, and DpR values as well as of HRs was set to 95% (*p* < 0.05). The ORR and DCR were compared between arms with Fisher’s exact test. All statistical analysis was performed with the use of R version 3.3.1 (The R Foundation for Statistical Computing, Vienna, Austria) and SAS version 9.4 (SAS Institute, Cary, NC, USA). A *p* value of <0.05 was considered statistically significant.

## 3. Results

### 3.1. Patients

A total of 211 patients was included in the initial trial database, with 49 patients being subsequently excluded from the study, 8 of 17 patients in HERBIS-2 and 41 of 110 patients in XParTS II because of the lack of a measurable lesion. All of the HERBIS-4A patients (n = 84) were included. IPD from a total of 162 patients were thus included in the current integrated analysis, with 79 patients in the SP arm and 83 patients in the XP arm (Appendix A). Baseline characteristics were well balanced between the study arms, with potential imbalances including fewer female patients (14.5% vs. 24.1%), gastric body tumors (30.1% vs. 41.8%), and differentiated tumor types (38.6% vs. 49.4%) in the XP arm compared with the SP arm (Table 1).

### 3.2. Survival

In the current integrated analysis, SP showed a better survival outcome over XP. Median PFS was thus more favorable for SP compared with XP at 5.9 months (95% CI, 4.5–7.4 months) versus 5.1 months (95% CI, 4.1–5.8 months), with an HR of 0.717 (95% CI, 0.512–1.005) and *p* value of 0.052 (Figure 1a). Median OS in the SP arm (14.2 months, 95% CI of 12.4–16.7 months) was significantly longer than that in the XP arm (11.0 months, 95% CI of 8.5–14.2 months), with an HR of 0.704 (95% CI, 0.500–0.989) and *p* value of 0.048 (Figure 1b).

#### ORR, Distribution of Tumor Shrinkage, and Changes in Tumor Burden

The ORR was similar between the SP and XP arms (48.1% vs. 50.6%, respectively, *p* = 0.755), whereas the DCR tended to be higher in the SP arm (83.5% vs. 72.3%, *p* = 0.137) (Appendix A). We also determined the DOR and TTR for patients who achieved a CR or PR (SP, n = 38; XP, n = 42). The median DOR for SP (7.0 months, 95% CI of 5.0–10.7 months) was double that for XP (3.5 months, 95% CI of 2.8–5.4 months), with this difference being statistically significant (*p* = 0.011) (Appendix A). On the other hand, the median TTR was similar between the SP and XP arms (2.1 months [95% CI, 1.8–2.3 months] vs. 1.8 months [95% CI, 1.6–2.2 months], respectively; HR of 0.741 [95% CI, 0.469–1.170]; *p* = 0.192) (Appendix A).

The distribution of tumor shrinkage for the SP and XP arms was visualized with waterfall plots (Figure 2a,b). Median DpR was 41.0% (95% CI, 15.4–64.0%) for the SP arm and 37.5% (95% CI, 8.47–53.5%) for the XP arm. Changes in tumor burden (assessed as the sum of the longest lesion dimensions) over time were visualized with spider plots (except for four and five cases of SP and XP arm, respectively, for which data were uncertain). Tumor reduction appeared to be more durable in patients who received SP (n = 75) (Figure 2c) than in those who received XP (n = 78) (Figure 2d), consistent with the significant differences in DOR between the two arms.

### 3.3. Differences in Treatment Effects between SP and XP According to Histology

We further investigated whether the treatment effects of SP and XP differed according to tumor histology. This analysis was performed for 77 patients in the SP arm (differentiated, n = 39; undifferentiated, n = 38) and 78 patients in the XP arm (differentiated, n = 32; undifferentiated, n = 46), excluding other histological types (2 and 5 cases in the SP and XP arms, respectively). We first examined differences in survival according to histology type. As seen in the overall analysis, there was no significant difference in median PFS between SP and XP arms for differentiated-type tumors (5.9 months [95% CI, 4.4–10.2 months] vs. 5.6 months [95% CI, 3.9–8.3 months], respectively; HR of 0.791 [95% CI, 0.479–1.307]; *p* = 0.358), whereas there was a trend for median PFS to be longer in the SP arm for undifferentiated-type tumors (5.9 months [95% CI, 4.4–10.2 months] vs. 4.4 months [95% CI, 4.1–5.8 months]; HR of 0.655 [95% CI, 0.404–1.061]; *p* = 0.086) (Figure 3a). A significant difference in median OS was apparent between the SP and XP arms for both tumor types, although the HR was lower for differentiated-type tumors (13.5 months [95% CI, 11.2–24.2 months] vs. 10.3 months [95% CI, 8.0–14.7 months]; HR of 0.506 [95% CI, 0.298–0.859]; *p* = 0.011) than for undifferentiated-type tumors (14.2 months [95% CI, 11.4–22.6 months] vs. 11.4 months [95% CI, 7.1–16.3 months]; HR of 0.790 [95% CI, 0.493–1.268]; *p* = 0.0011) (Figure 3b). The difference in OS between SP and XP arms was thus greater than that in PFS for differentiated tumors, whereas differences in PFS between the two arms appeared to consistently reflect that in OS for undifferentiated tumors.

### 3.4. Association of the Longer OS in the SP Arm with Deeper Tumor Shrinkage for Differentiated Tumors

We further investigated whether the survival differences between SP and XP arms dependent on tumor histology were due to differences in tumor response rate or shrinkage, the latter analysis being performed for 73 patients in the SP arm (differentiated, n = 37; undifferentiated, n = 36) and 76 patients in the XP arm (differentiated, n = 31; undifferentiated, n = 45), excluding for four and five cases of SP and XP arm, respectively, for which data were uncertain. The ORR and DCR tended to be higher for differentiated tumors (ORR, 53.8% for SP and 56.3% for XP, *p* = >0.999; DCR, 87.2% for SP and 78.1% for XP, *p* = 0.354) than for undifferentiated tumors (ORR, 44.7% for SP and 45.7% for XP, *p* = >0.999; DCR, 81.6% for SP and 69.6% for XP, *p* = 0.311), with no differences between the treatment arms. However, deeper tumor shrinkage was apparent in the SP arm than in the XP arm for differentiated tumor types (Figure 4a,b), as shown by the fact that cases that achieved a >60% reduction were significantly more common in the SP arm than in the XP arm (29.7% [11/37] vs. 6.4% [2/31]; *p* = 0.038, Fisher’s exact test), suggesting that this deeper shrinkage may have contributed to the longer OS for the SP arm. In contrast, DpR appeared similar between SP and XP arms for undifferentiated tumor types (Figure 4c,d), but patients whose tumors increased in size from baseline tended to be less frequent in the SP arm than in the XP arm (2.8% [1/36] vs. 13.3% [6/45]; *p* = 0.123, Fisher’s exact test). Spider plots for each treatment arm according to histological type (Figure 4e–h) showed that a transient response in the XP arm was apparent for both tumor types, but was more pronounced for undifferentiated tumors, reflecting the shorter PFS in the XP arm than in the SP arm for this subgroup. Overall, these findings thus suggested that the therapeutic benefit of SP over XP as first-line treatment varies greatly depending on histological type.

Finally, we performed subgroup analysis for PFS and OS according to tumor histology (differentiated type vs. undifferentiated type) and tumor shrinkage (tumor size reduction of ≥30% from baseline vs. <30%) (Figure 5). Although both PFS and OS tended to be better for SP than for XP in most subgroups, a significant benefit of SP versus XP was apparent only for OS in patients with differentiated tumors and a tumor size reduction of ≥30% (23.7 months [95% CI, 13.2–not available] vs. 11.7 months [95% CI, 7.8–19.6 months]; HR of 0.339 [95% CI, 0.163–0.705]; interaction *p*  =  0.003; Appendix A), suggesting that deeper tumor shrinkage induced by SP compared with XP contributed to prolongation of OS for patients with differentiated tumors.

## 4. Discussion

There have been various comparisons between S-1–based and capecitabine-based regimens for first-line treatment of advanced gastric cancer, but none of the individual trials [10,11,12] or combined abstract-based analyses [13,14,15,16,17,18] has shown a significant survival difference. The present analysis showed that SP significantly outperformed XP in terms of OS, with a trend toward a better PFS also for the SP arm. As far as we are aware, this is the first evidence of a significant OS benefit for SP over XP. This significant finding may have emerged as a result both of the increased number of patients in our combined analysis of the three original trials and of the fact that the analysis was limited to patients with measurable disease. In addition, it is important to note that the present analysis was based on IPD rather than abstract based, with the latter approach evaluating only median values. On the other hand, there was no significant difference in the ORR between SP and XP. Given that the ORR corresponds to the percentage of patients whose tumors shrink by ≥30%, which is commonly adopted as an indicator of treatment efficacy, it serves as a “temporary measure” during treatment and may not reveal efficacy over the entire treatment period. We found that the pattern of tumor shrinkage differed between the SP and XP arms, with spider plots clarifying that this difference likely accounts for the significant difference in DOR between the two arms. A high DCR and durability of tumor shrinkage are advantageous for gastric cancer patients, given that worsening tumor control is directly related to loss of quality of life as a result of the inability to eat.

Despite the difference in drug design, it has remained uncertain how to discriminate the use of S-1 from that of capecitabine. We have now shown that these two drugs differ qualitatively in terms of efficacy in a manner dependent on tumor histology. Of interest, no difference in the ORR was apparent between SP and XP for subgroups based on tumor histology. However, for undifferentiated tumors, the percentage of patients whose tumors did not shrink tended to be greater for XP than for SP. This difference may directly reflect the observed differences in PFS, TTF, and OS. On the other hand, for differentiated tumors, SP resulted in deeper tumor shrinkage compared with XP, as shown by the observation that the proportion of patients who achieved a tumor reduction of >60% was significantly greater for SP. The cutoff value for DpR of 60% seemed reasonable given that cutoff values of 50% or 62.4% were previously adopted as predictors of longer survival for first-line chemotherapy in Japanese patients with HER2-positive gastric cancer, for whom the median DpR was 56.8% [19], and in patients with colorectal cancer [20], respectively. Our subgroup analysis indicated that deeper tumor shrinkage induced by SP compared with XP contributed to OS prolongation in patients with differentiated tumors. Association of no difference in ORR and PFS with a large difference in OS, as observed for differentiated tumors in the present study, can occur when DpR is large, as has been seen in previous studies [21,22] and given that an increased DpR is more associated with prolongation of postprogression survival and therefore OS than with the ORR or PFS [21,22]. Together, our data have thus revealed that qualitative differences between the effects of SP and XP were highlighted by tumor histological classification.

Our study has several limitations. Although it was based on IPD from prospective studies, it was conducted in an ad hoc setting and should be interpreted with caution. In addition, our analysis was limited to patients with measurable lesions, and the results may therefore not be applicable to all patients. Furthermore, the platinum agent of the treatment regimens was cisplatin, and so it is not clear whether our results will also be applicable to regimens based on other platinum agents. In recent years, oxaliplatin has become more commonly administered than cisplatin as the platinum agent in combination with fluoropyrimidines for gastric cancer. Indeed, S-1 plus oxaliplatin (SOX) and capecitabine plus oxaliplatin (CapeOX) are currently recommended regimens in the treatment guidelines. A previous randomized phase II study showed no significant difference between SOX and CapeOX in efficacy such as OS, PFS, and ORR [23], but they lacked the information regarding DpR or histology type as an assessment of efficacy. Currently, nivolumab, an immune checkpoint inhibitor, in combination with oxaliplatin and fluoropyrimidines is a standard therapy for HER2-negative unresectable advanced or recurrent gastric cancer on the basis of recent phase III studies [24,25], raising the issue of whether SOX or CapeOX should be used as backbone chemotherapy. Of interest, in the phase II part of the latter (ATTRACTION-4) study [26], although the ORR was similar for the SOX-nivolumab arm and the CapeOX-nivolumab arm (66.7% vs. 70.6%, respectively), the waterfall plot is suggestive of a deeper tumor shrinkage in the SOX arm than in the CapeOX arm, consistent with our results. Further studies focusing on histology are needed to clarify the difference in efficacy of SOX versus CapeOX in combination with nivolumab, given that a recent study suggested an association between a high tumor burden and immunosuppressive phenotypes [27].

## 5. Conclusions

In conclusion, our data have shown that SP is superior to XP for patients with HER2-negative unresectable advanced or recurrent gastric cancer, but that there are qualitative differences between the effects of SP and XP that depend on the histological type of the tumor. For undifferentiated tumors, SP has fewer treatment failures than does XP, as is reflected in a better PFS, and OS. For differentiated tumors, SP achieves deeper tumor shrinkage compared with XP, which contributes to a longer OS, but not PFS. Further study is warranted to determine whether these differences for S-1 versus capecitabine are reproduced in combination with oxaliplatin and immune checkpoint inhibitors, the new standard of care for HER2-negative unresectable advanced or recurrent gastric cancer.

## Figures and Tables

**Figure 1 cancers-14-05673-f001:**
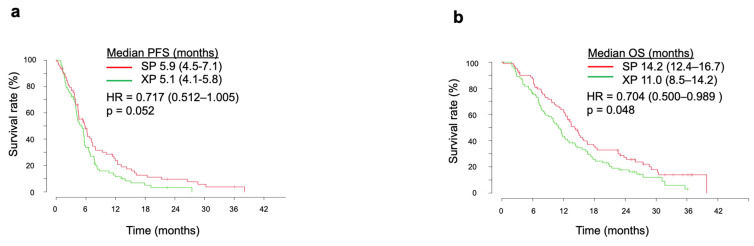
Kaplan–Meier analysis of PFS (**a**), and OS (**b**) in the SP and XP arms in the integrated analysis. The *p* values were determined with the log-rank test.

**Figure 2 cancers-14-05673-f002:**
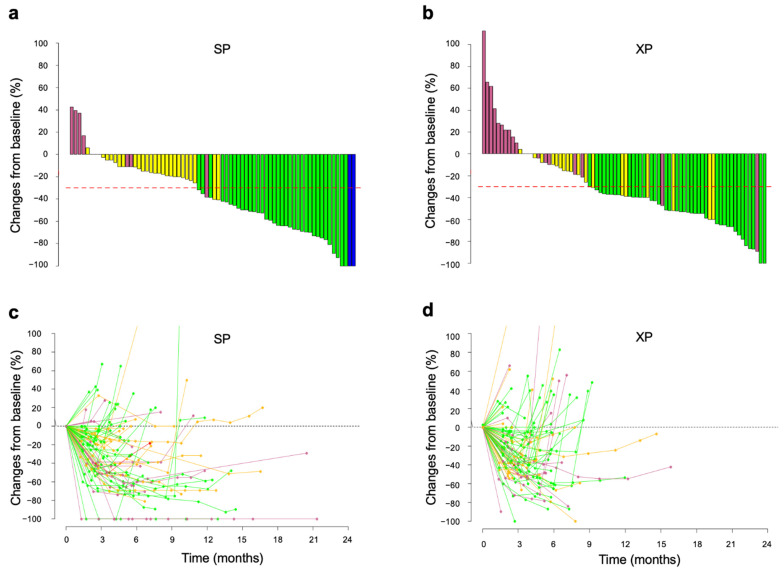
Changes in tumor size during treatment with SP or XP. (**a**,**b**) Best change from baseline in the sum of the longest target lesion diameters for each patient receiving SP (**a**) or XP (**b**). Best responses are color-coded: blue, CR; green, PR; yellow, SD; pink, PD. Red dashed lines indicate tumor shrinkage of 30%. (**c**,**d**) Time course of the percentage change in the sum of the longest target lesion diameters from baseline in each patient receiving SP (**c**) or XP (**d**).

**Figure 3 cancers-14-05673-f003:**
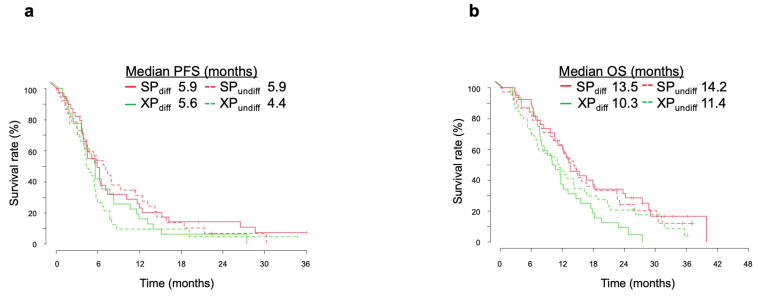
Kaplan–Meier analysis of PFS (**a**), and OS (**b**) according to differentiated (diff, solid lines) or undifferentiated (undiff, dotted lines) tumor types in the SP and XP arms of the integrated analysis.

**Figure 4 cancers-14-05673-f004:**
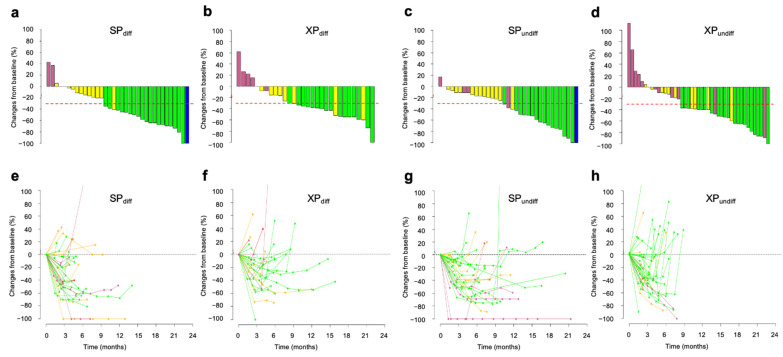
Changes in tumor size according to tumor histology during treatment with SP or XP. (**a**–**d**) Best change from baseline in the sum of the longest target lesion diameters for differentiated tumors (diff, **a**,**b**) or undifferentiated tumors (undiff, **c**,**d**) in each patient treated with SP (**a**,**c**) or XP (**b**,**d**). Best responses are color-coded: blue, CR; green, PR; yellow, SD; pink, PD. Red dashed lines indicate tumor shrinkage of 30%. (**e**–**h**) Time course of the percentage change in the sum of the longest target lesion diameters from baseline for differentiated tumors (diff, **e**,**f**) or undifferentiated tumors (undiff, **g**,**h**) in each patient treated with SP (**e**,**g**) or XP (**f**,**h**).

**Figure 5 cancers-14-05673-f005:**
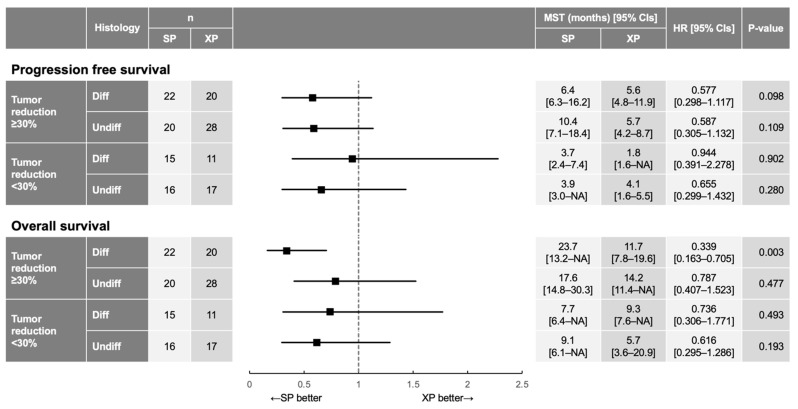
HR and 95% CI for PFS and OS in patient subgroups defined by the extent of tumor reduction and tumor histology. The *p* values were determined with the log-rank test. MST, median survival time; NA, not available.

**Table 1 cancers-14-05673-t001:** Baseline characteristics of patients in the SP and XP arms for the integrated analysis.

	SP				XP			
	All (*n* = 79)	HERBIS-2 (*n* = 5)	HERBIS-4A (*n* = 41)	XParTS II (*n* = 33)	All (*n* = 83)	HERBIS-2 (*n* = 4)	HERBIS-4A (*n* = 43)	XParTS II (*n* = 36)
Age (years)								
Median [min, max]	66 [37, 77]	71 [64, 73]	68 [37, 77]	65 [44, 74]	65 [31, 79]	73 [72, 74]	64 [34, 79]	65 [31, 74]
Sex								
Male	60 (75.9)	5 (100.0)	33 (80.5)	22 (66.7)	71 (85.5)	4 (100.0)	36 (83.7)	31 (86.1)
Female	19 (24.1)	0 (0.0)	8 (19.5)	11 (33.3)	12 (14.5)	0 (0.0)	7 (16.3)	5 (13.9)
ECOG PS								
0	54 (68.4)	4 (80.0)	22 (53.7)	28 (84.8)	59 (71.1)	4 (100.0)	24 (55.8)	31 (86.1)
1	24 (30.4)	1 (20.0)	19 (46.3)	4 (12.1)	23 (27.7)	0 (0.0)	19 (44.2)	4 (11.1)
2	1 (1.3)	0 (0.0)	0 (0.0)	1 (3.0)	1 (1.2)	0 (0.0)	0 (0.0)	1 (2.8)
HER2 status								
Positive	0 (0.0)	0 (0.0)	0 (0.0)	0 (0.0)	0 (0.0)	0 (0.0)	0 (0.0)	0 (0.0)
Negative	76 (96.2)	4 (80.0)	41 (100.0)	31 (93.9)	79 (95.2)	4 (100.0)	42 (97.7)	33 (91.7)
Unknown	3 (3.8)	1 (20.0)	0 (0.0)	2 (6.1)	4 (4.8)	0 (0.0)	1 (2.3)	3 (8.3)
Tumor localization								
U	21 (26.6)	2 (40.0)	7 (17.1)	12 (36.4)	26 (31.3)	2 (50.0)	14 (32.6)	10 (27.8)
M	33 (41.8)	2 (40.0)	21 (51.2)	10 (30.3)	25 (30.1)	0 (0.0)	11 (25.6)	14 (38.9)
L	22 (27.8)	1 (20.0)	13 (31.7)	8 (24.2)	31 (37.3)	2 (50.0)	18 (41.9)	11 (30.6)
E	1 (1.3)	0 (0.0)	0 (0.0)	1 (3.0)	0 (0.0)	0 (0.0)	0 (0.0)	0 (0.0)
D	2 (2.5)	0 (0.0)	0 (0.0)	2 (6.1)	1 (1.2)	0 (0.0)	0 (0.0)	1 (2.8)
Histology								
pap	0 (0.0)	0 (0.0)	0 (0.0)	0 (0.0)	3 (3.6)	0 (0.0)	2 (4.7)	1 (2.8)
tub1	10 (12.7)	0 (0.0)	8 (19.5)	2 (6.1)	6 (7.2)	1 (25.0)	4 (9.3)	1 (2.8)
tub2	29 (36.7)	2 (40.0)	15 (36.6)	12 (36.4)	23 (27.7)	1 (25.0)	15 (34.9)	7 (19.4)
por1	22 (27.8)	0 (0.0)	10 (24.4)	12 (36.4)	22 (26.5)	0 (0.0)	9 (20.9)	13 (36.1)
por2	10 (12.7)	2 (40.0)	3 (7.3)	5 (15.2)	15 (18.1)	0 (0.0)	9 (20.9)	6 (16.7)
sig	4 (5.1)	0 (0.0)	3 (7.3)	1 (3.0)	7 (8.4)	1 (25.0)	0 (0.0)	6 (16.7)
muc	2 (2.5)	1 (20.0)	0 (0.0)	1 (3.0)	2 (2.4)	1 (25.0)	1 (2.3)	0 (0.0)
Other	2 (2.5)	0 (0.0)	2 (4.9)	0 (0.0)	5 (6.0)	0 (0.0)	3 (7.0)	2 (5.6)
TNM classification								
M stage								
MX	2 (2.5)	0 (0.0)	2 (4.9)	0 (0.0)	2 (2.4)	0 (0.0)	1 (2.3)	1 (2.9)
M0	17 (21.5)	5 (100.0)	6 (14.6)	6 (18.2)	15 (18.3)	4 (100.0)	4 (9.3)	7 (20.0)
M1	60 (75.9)	0 (0.0)	33 (80.5)	27 (81.8)	65 (79.3)	0 (0.0)	38 (88.4)	27 (77.1)
Prior surgery								
No	62 (78.5)	0 (0.0)	34 (82.9)	28 (84.8)	68 (81.9)	0 (0.0)	39 (90.7)	29 (80.6)
Yes	17 (21.5)	5 (100.0)	7 (17.1)	5 (15.2)	15 (18.1)	4 (100.0)	4 (9.3)	7 (19.4)

All entries are number (%) with the exception of age. Abbreviations not defined in text: ECOG PS, Eastern Cooperative Oncology Group performance status.

## Data Availability

The data that support the findings of this study are available from the corresponding author, H.K., upon reasonable request.

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
