# Peer review of "Histology Classification Highlights Differences in Efficacy of S-1 versus Capecitabine, in Combination with Cisplatin, for HER2-Negative Unresectable Advanced or Recurrent Gastric Cancer with Measurable Disease"

_cancers, 2022, doi:10.3390/cancers14225673_

Round 1
Reviewer 1 Report
Currently, S-1 or capecitabine combined with platinum agents is the preferred first-line treatment for unresectable advanced or recurrent HER2-negative GC with measurable disease. It is unclear, however, which is the preferred combination with platinum agents.
Interestingly, the authors demonstrated that SP (S-1 [40–60 mg twice daily for 21 days] plus cisplatin [60 mg/m2 on day 8], 40 every 5 weeks) is superior to XP (capecitabine [1000 mg/m2 twice daily for 14 days] plus cisplatin [80 mg/m2 41 on day 1], every 3 weeks) for patients with HER2- negative unresectable advanced or recurrent gastric cancer. According to the findings, there are qualitative differences between the effects of SP and XP that depend on the histological type of the tumor. For undifferentiated tumors, SP has fewer treatment failures than does XP, as is reflected in a better PFS, and OS. For differentiated tumors, SP achieves deeper tumor shrinkage compared with XP, which contributes to a longer OS, but not PFS.
These findings are interesting and worth publishable. However, I have two concerns.
1. As far as tumor histology is concerned, the author did not produce any figures illustrating the status of the tumors in patients. For the researchers, it would be helpful if the authors could provide representative tumor histological figures that they have used for histological evaluation of tumor shrinkage.
2. There is also a concern regarding the discussion. It would be helpful if authors explained why this combination of SP and XP is more effective in HER2- tumors than in HER2+ tumors. Please include such details in the discussion.
Author Response
We thank the reviewer for insightful and positive comments, which have helped us improve our manuscript. Our specific responses to the points raised are as follows:
- As far as tumor histology is concerned, the author did not produce any figures illustrating the status of the tumors in patients. For the researchers, it would be helpful if the authors could provide representative tumor histological figures that they have used for histological evaluation of tumor shrinkage.
We agree with the reviewer’s comment that showing representative figures of differentiated and undifferentiated tumors would be helpful for readers. Unfortunately, all three studies used in this integrated analysis are conducted nearly 10 years ago, and the limited revision period does not allow us to prepare histopathology images of those cases. We believe that studies based on the histological classification of differentiated vs. undifferentiated types are common in gastric cancer and are usually understood by readers even without a representative figure.
- There is also a concern regarding the discussion. It would be helpful if authors explained why this combination of SP and XP is more effective in HER2- tumors than in HER2+ tumors. Please include such details in the discussion.
Thanks for the reviewer’s insightful comment. We are aware that only HER2-negative gastric cancer has been tested in comparison with SP and XP, but not in HER2-positive gastric cancer, ToGA trial (Bang YJ, et al. Lancet. 2010 Aug 28;376(9742):687-97) showed that XP plus trastuzumab is the standard of care, whereas SP can be used as an alternative to XP as demonstrated by a single-arm phase II. As Reviewer raised, there should be a comparison between S-1 and capecitabine in HER2-positive gastric cancer as well.
Reviewer 2 Report
This study demonstrates that SP is superior to XP for patients with HER2- negative unresectable advanced or recurrent gastric cancer, but that there are qualitative differences between the effects of SP and XP that depend on the histological type of the tumor. For undifferentiated tumors, SP has fewer treatment failures than does XP, as is reflected in a better PFS, and OS. For differentiated tumors, SP achieves deeper tumor shrinkage compared with XP, which contributes to a longer OS, but not PFS. The result is important for clinical guidance. I think it can published after the following comments are addressed.
1. For oral fluoropyrimidine drugs. Is there any report that using Intravenous injection.
2. Based on the results, what is the author's opinion for the clinic application.
3. For survival rate, the number of the group is missing.
Author Response
We thank the reviewer for insightful and positive comments, which we feel have helped us to improve our manuscript. Our specific responses to the points raised are as follows:
- For oral fluoropyrimidine drugs. Is there any report that using Intravenous injection.
The JCOG9912 study (Boku N, et al. Lancet Oncol. 2009 Nov;10(11):1063-9) conducted in Japan and the ML17032 study (Kang YK, et al. Ann Oncol. 2009 Apr;20(4):666-73) compare oral FU agents with injectable FU agents. The former proved non-inferiority of S-1 to injectable 5-FU, and the latter proved non-inferiority of capecitabine+cisplatin to 5-FU+cisplatin. Neither study focused on histology and quality of efficacy as we did here.
- Based on the results, what is the author's opinion for the clinic application.
We thank the reviewer’s asking a very important question. As we demonstrated in this manuscript, S-1 shows consistently better clinical outcome over capecitabine both in differentiated and undifferentiated tumors, with the former being characterized by a deeper tumor shrinkage. Currently, S-1 or capecitabine plus oxaliplatin is adopted as a backbone chemotherapy for nivolumab in the 1st line setting against HER2 negative gastric cancer. In eliciting therapeutic benefit from nivolumab, deeper tumor shrinkage is important, and thus we recommend utilizing S-1, instead of capecitabine for HER2 negative gastric cancer.
- For survival rate, the number of the group is missing.
We thank the reviewer’s careful reading. In our analysis of tumor histology, the number of cases was incorrect for the analysis of distribution of tumor shrinkage. The correct numbers are as follows: the tumor shrinkage analysis was performed for 73 patients in the SP arm (differentiated, n = 37; undifferentiated, n = 36) and 76 patients in the XP arm (differentiated, n = 31; undifferentiated, n = 45), excluding for four and five cases of SP and XP arm, respectively, for which data were uncertain. We have now corrected these numbers and relevant information in the revised manuscript.